



# Biomass Yield Potential, Feedstock Quality, and Nutrient Removal of Perennial Buffer Strips under Continuous Zero Fertilizer Application

Cheng-Hsien Lin[1,2], Colleen Zumpf[3], Chunhwa Jang[2], Thomas Voigt[2], Guanglong Tian[4], Olawale Oladeji[4], Albert Cox[4], Rehnuma Mehzabin[2], and DoKyoung (D.K.) Lee[2,3*]

[1]Department of Soil and Environmental Sciences, National Chung Hsing University, Taichung 40227, Taiwan
[2]Department of Crop Sciences, University of Illinois at Urbana-Champaign, Urbana, Illinois 61801, USA
[3]Argonne National Laboratory, Environmental Science Division, Illinois 60439, USA
[4]Monitoring and Research Department, Metropolitan Water Reclamation District of Greater Chicago, Cicero, Illinois 60804, USA

*Correspondence to*: DoKyoung (D.K.) Lee (leedk@illinois.edu)

**Abstract.** Perennial-based buffer strips have been promoted with the potential to improve ecosystem services from riparian areas while producing biomass as livestock feed or as a bioenergy feedstock. Both biomass production and nutrient removal of buffer strips are substantially influenced by the vegetation types for the multipurpose perennial buffers. In this 2016-2019 study in Western Illinois, two perennial cropping systems, including forage crops composed of cool-season grass mixtures

(forage system) and bioenergy crops made up of warm-season grass mixtures (bioenergy system), were used to establish buffer strips for assessing biomass production, feedstock quality, nutrient removals, and buffer longevity. Treatments for this study reflecting agronomic practices included 1) two harvests occurring in summer (at anthesis) and fall (after complete senescence) and 2) one harvest in fall for forage system (2- vs. 1-cut forage), and 3) one fall harvest for bioenergy system (1-cut bioenergy). Successively harvesting without any fertilizer input resulted in a yield decline in forage biomass over three years by

approximately 30% (6.3 to 4.4 DM Mg ha$^{-1}$ with the rate of 1.0 Mg ha$^{-1}$ yr$^{-1}$) in the 2-cut forage and by 35% (4.9 to 3.2 DM Mg ha$^{-1}$ with the rate of 0.9 Mg ha$^{-1}$ yr$^{-1}$) in the 1-cut forage systems. The feed quality also decreased over the years by showing declined rates of 12.9 (crucial protein), 0.9 (calcium), 0.7 (copper), and 1.3 (zinc) g kg$^{-1}$ DM yr$^{-1}$. Empirical models predicted enteric $CH_4$ emissions from cattle ranged from 225.7 to 242.6 g cow$^{-1}$ d$^{-1}$ based on the feed nutritive values. In contrast, bioenergy biomass yield increased by 27% from 4.9 to 6.7 DM Mg ha$^{-1}$ with consistent quality (cellulose ~ 397.9 g kg$^{-1}$;

hemicellulose ~ 299.4 g kg$^{-1}$), corresponding to the increased total theoretical ethanol yield from $1.8 \times 10^3$ to $2.4 \times 10^3$ L ha$^{-1}$ (~33% increase). Annual nutrient removals of N, P, K, Ca, and Mg were significantly higher in the forage systems (e.g., 2-cut: 52.6~106.9 kg-N ha$^{-1}$; 1-cut: 44.5~84.1 kg-N ha$^{-1}$) than those in the bioenergy system (e.g., 25.9~34.4 kg-N ha$^{-1}$); however, the removal rate declined rapidly over three years (e.g., ~49% reduction) as the annual biomass yield declined in the forage systems. This on-farm field study demonstrated the potential of the perennial crop used as buffer strip options for biomass

production and buffer sustainability at the edge of the field.



## 1 Introduction

The edges of cropped lands are usually less productive/profitable for conventional row crop cultivation due to rapid nutrient transportation via surface runoff and leaching (Dodds and Oakes, 2008; David et al., 2010). Field borders are hotspots for severe environmental degradation where substantial erosion and nutrient loss to adjacent water bodies can result in poor soil and water quality and contribute to downstream impacts such as the Gulf of Mexico Hypoxic Zone (Turner and Rabalais, 1994; Rabalais et al., 2002; Vitousek et al., 2009; Márquez et al., 2017). Planting buffer strips along field edges can effectively combat these problems by preventing sediment transportation and intercepting/removing excess nutrients (Lovell and Sullivan, 2006; Sweeney and Newbold, 2014; Schmitt et al., 1999; Gopalakrishnan et al., 2012). Perennial grasses such as tall fescue (cool-season/C3 forage grasses), switchgrass, and Miscanthus (warm-season/C4 bioenergy grasses) species are suggested as good candidates for buffer strips due to their high stress tolerances and efficient take-up abilities (Dosskey, 2001; Clausen et al., 2000; Vogel at al., 2002; Mulkey et al., 2006; Varvel et al., 2008; van der Weijde et al., 2013). These perennials have shown great potential to provide numerous ecosystems services (e.g., the mitigation of soil erosion and greenhouse gas emissions, the improvement of soil health and biodiversity, soil carbon sequestration, etc.) for marginal areas (Lee et al., 2007; Monti et al., 2012; Yang et al., 2019). Also, a productive perennial system may provide landowners with alternative income sources through the sale of harvested biomass as either forage or bioenergy feedstocks (Lee et al., 2018; Mehmood et al., 2017; Eranki et al., 2013; Anderson et al., 2016; Golkowska et al., 2016).

A minimum requirement of a successful perennial buffer is to offer stable yield production with high-quality feedstock. The key is to establish a persistent buffer system to serve the agroecosystem of marginal croplands continuously. For perennial monocultures, a successively productive system can be achieved under appropriate management. For instance, well-managed alfalfa and switchgrass can last 5-10 years or longer under proper nutrient and harvest management (Bélanger et al., 2006; Parrish and Fike 2005; Fike et al. 2006a; Lee et al., 2018). Harvest management, mainly the harvest frequency (e.g., single-cut vs. multiple-cut) and timing (e.g., at anthesis/peak standing vs. at the end of growing season/after killing frost), can be identified as the most critical factors to influence the perennial's health and production (Guretzky et al., 2011). Frequent harvesting could improve the annual biomass yield, especially for cool-season grasses that typically have two growth cycles (spring and late-summer/autumn) per year (MacAdam and Nelson, 2003); however, intensive harvest (e.g., >3 harvests per year) likely reduces perennials regrowth vigor compared to the less intensive harvest management (e.g., 2 harvests per year) (Bélanger et al., 2006). For warm-season grasses, a single annual harvest has been suggested to optimize biomass yield, minimize energy inputs, and maintain stand persistence (Sanderson et al., 1999; Mitchell and Schmer, 2012). Harvest timing is another key factor to ensure a sustainable perennial system. Many studies show the delayed harvest after plant senescence substantially improves stand regrowth potentials and longevity by increasing internal nutrient cycling and lengthening the time for root development and productive tiller growth (Mitchell and Schmer, 2012; Vogel et al., 2002; Lee et al., 2014; Zumpf et al., 2019).



Compared to monoculture systems, polycultures showed more advantages in terms of ecosystem services, resistances to weed and pest pressure, and biomass productivity (Carlsson et al., 2017; De Deyn et al., 2011; Dhakal & Islam, 2018; Jungers et al., 2015; Nyfeler et al., 2011; Quijas et al., 2010; Sanderson et al., 2012; Suter et al., 2015; Yang et al., 2019; Lin et al., 2022). Nevertheless, it is more complicated and challenging to optimize management practices in polycultures (e.g., grass mixtures) than in monoculture systems for simultaneously achieving stable productivity, feedstock quality, and system longevity because each species responds differently to different treatments. For instance, studies have observed increasing nitrogen (N) inputs usually benefited biomass yield of perennial grasses but not so much for legume production and persistence (Harmoney et al., 2016; Lee et al., 2013; Lin et al., 2022). From a quality standpoint, additional N input improved the quality of forage feedstocks (e.g., cool-season grasses) by increasing protein content in plant tissue but lowered the quality of lignocellulosic feedstocks (e.g., warm-season grasses) by reducing the concentration of cell wall components and increasing the biomass-N and ash contents for bioenergy production (Hodgson et al., 2010. Ibrahim et al., 2017; Lin et al., 2022). For harvest management, our previous studies showed that delayed harvest after complete senescence and less frequent harvest practices can improve the sustainability of the legume-grass polycultures established on Conservation Reserve Program (CRP) lands, in terms of the long-term productivity and vegetative vigor (Lin et al., 2022).

Perennial-plant buffers designed for multifunctionality offer opportunities to bridge biomass production and ecosystem services in agricultural watersheds, however, limited regional information has identified optimal plant types coupled with specific practices for producers (Lovell and Sullivan, 2006; Golkowska et al., 2016). Research on site-specific and continuous cropping practices can provide local farmers with valuable information to develop strategic plans for improving local economics and the sustainability of buffer systems, simultaneously. The overall goals of this study were to evaluate the effect of minimum management efforts (only harvest without nutrient applications) on yield potential, feedstock quality, nutrient removal, and buffer sustainability of cool- and warm-season perennial mixtures cultivated on low-fertile marginal croplands. Since no additional fertilizer was applied to the buffer strips, the nutrients from parent soil materials and transported from the main crop fields via leaching and surface runoff were the only nutrient sources for the perennials. Thus, it was hypothesized that the warm-season perennial buffer (bioenergy system) could be more sustainable than the cool-season perennials (forage system) due to its high nutrient use efficiency, great ability of nutrient scavenging, and stress tolerance for the marginal area (van der Weijde et al., 2013; Pedroso et al., 2014; Lee et al., 2018). Besides the evaluation of the continuous biomass supply of both forage and bioenergy-type buffers under local common harvest practices, the specific objectives of this study were to assess their feedstock quality, including 1) nutritive values of crude protein (CP), crude fibers of neutral detergent fiber (NDF), acid detergent fiber (ADF), the digestibility-related indices, and other essential elements for ruminants feeds, 2) dynamics of cell wall compositions (e.g., cellulose, hemicellulose, and lignin) for lignocellulosic biofuel productions, and 3) to predict the enteric methane ($CH_4$) production based these nutritive qualities.



## 2 Materials and Methods

### 2.1 Site description

This study was conducted during the 2016-2019 growing seasons in a riparian area of a 10-ha field located in Fulton County, Illinois (IL, 40°28'23" N; 90°6'44" W) with an annual corn-soybean (C-S) rotation cropping history (Fig. 1). The site is in a temperate climate region with a 30-year average annual temperature of 11°C and annual precipitation of 970 mm. Weather information, including monthly temperature and precipitation, and cumulative precipitation, from 2016 to 2019, along with 30-year averages (1990-2019) were obtained from the National Oceanic and Atmospheric Administration for Fulton County, IL (Peoria International Airport station, USW00014842) and shown in Figure 2. The field is mostly a Sawmill silty clay loam soil (Fine-silty, mixed, superactive, mesic Cumulic Endoaquolls), while Wakeland silt loam (Coarse-silty, mixed, superactive, nonacid, mesic Aeric Fluvaquents) is found at its southern edge along Big Creek (Soil survey, USDA, 2016). Nine soil cores (0-100 cm) were collected across the riparian area in 2016, and each core was segmented into 0-5, 5-10, 10-20, 20-30, 30-60, and 60-100 cm depths and analyzed for more soil details (Table 1).

### 2.2 Experimental design

Two crop systems (forage and bioenergy) and two harvest frequencies (2- or 1-cut annually) were used in the experiment. The forage-crop system (forage system) was composed of alfalfa (*Medicago sativa* L.) at a seeding rate of 5.6 kg ha$^{-1}$, and cool-season grasses at a total seeding rate of 28 kg ha$^{-1}$, which included smooth bromegrass (*Bromus inermis Leyss.*, 20%), tall fescue (*Lolium arundinaceum [Schreb.] S.J. Darbyshire*, 20%), orchardgrass (*Dactylis glomerata L.*, 20%), perennial ryegrass (*Lolium perenne* L. ssp. perenne, 15%), Timothy (*Phleum pratense* L., 15%), and meadow fescue (*Schedonorus pratensis [Huds.] P. Beauv.*, 10%). The bioenergy-crop system (bioenergy system) was composed of warm-season grasses at a seeding rate of 12 kg ha$^{-1}$, including switchgrass (*Panicum virgatum* L., 40%), big bluestem (*Andropogon gerardii Vitman*, 20%), Indiangrass (*Sorghastrum nutans* [L.] Nash, 20%), and prairie cordgrass (*Spartina pectinata* Link, 20%). The seeding rate of both cool- and warm-season grass mixtures was 323 pure live seed (PLS) m$^{-2}$. Based on the grass growing behavior, the forage system was either harvested twice, at anthesis in early summer (June) and at the end of the growing season (after complete senescence) in the fall (2-cut), or once in the fall (1-cut) annually (October - November). The bioenergy system was only harvested in the fall (Table 2). Thus, three treatments, including two forage (2-cut vs. 1-cut) and one bioenergy (1-cut) systems, were used in this experiment within a randomized complete block with three replicates at each location. For each treatment, the plot size was approximately 385 m$^2$.

### 2.3 Field management

The buffer strips were established in May of 2016. Disking was conducted for seedbed preparation and herbicide was used to control weeds prior to planting. Glyphosate was applied in the forage systems, and atrazine was applied in the bioenergy system. The seeds were directly drilled into a firm, non-tilled seedbed at approximately 10 mm deep with a row spacing of 15





cm using a Great Plains no-till drill (Salina, KS). Replanting was done for the bioenergy system in April of 2017 due to the substantial loss of grasses resulting from the herbicide drift from the C-S field. Before the replanting, the plots were burned to remove dead biomass residues and atrazine was applied two weeks later to control weeds. For the fertilizer management in the

C-S field aqueous urea ammonium nitrate (UAN: 32-0-0) was used as the N source, and 202 kg N ha$^{-1}$ was applied before corn planting. Diammonium phosphate (DAP: 18-46-0) was applied to soybean at the rate of 28 kg P$_2$O$_5$ ha$^{-1}$ before planting. No fertilizers were applied to the buffer strips of either perennial system. The first harvest year was different between the two crop systems. For the forage system, the first harvest occurred in 2017 because of the insufficient biomass in the establishment year of 2016. For the bioenergy system, the first harvest year was delayed to 2018 due to the reestablishment in 2017.

**2.4 Data collection, analysis, and calculations**

Aboveground biomass from a 14 m × 2.8 m area in each forage and bioenergy system was harvested at a height of 10 cm using a biomass plot harvester (Cibus S, Wintersteiger, Salt Lake City, UT). The fresh weight of the harvested biomass from each plot was measured, and the dry matter (DM) weight was determined by placing wet-matter subsamples (~ 1 kg) in a forced-air oven at 60˚C for five days. The oven-dried biomass was then weighed and ground to pass through a 1-mm screen using a

Retsch cutting mill (Retsch Inc., Haan, Germany) for feedstock quality analysis and biomass nutrients. Fiber analyses were analyzed by measuring the concentrations of neutral detergent fiber (NDF), acid detergent fiber (ADF), and acid detergent lignin (ADL) which were analyzed using a sequential extraction and filtration process (Ankom Technology, 2002 and 2003) and an Ankom 200 Fiber Analyzer (ANKOM Technology, Fairport, NY, USA). Plant-tissue ash content was determined from the mass lost by placing the dry sample in a muffle furnace at 600°C for 8 hours. For the feedstock nutrient analysis, the dried

biomass was analyzed by a dry combustion method using a LECO FP-528 N Determinator (Leco Inc., St. Joseph, MI). Other macro- and micro-nutrients were measured using inductively coupled plasma spectrometry with optical emission spectrometry (Thermo Scientific iCAP 6500 Duo ICP, Thermo Fisher Scientific Inc., Waltham, MA) following a concentrated nitrate acid and hydrochloric acid microwave digestion procedure in a MARSXpress vessel (CEM, Matthews, NC).

Feedstock quality indices were calculated on a dry matter basis using the measured fiber compositions (i.e., NDL, ADF, and ADL) and equations shown in Table 3 (Ameen et al., 2019). Quality Indices of animal feed included crude protein (CP), dry matter intake (DMI), dry matter digestibility (DMD), total digestible nutrients (TDN), net energy for lactation (NEL), and relative feed value (RFV). For bioenergy productions based on the biochemical process, the concentrations of cellulose and hemicellulose (Hemi-C), theoretical ethanol yield (TEY), and total theoretical ethanol yield (TTEY) are commonly used

indices. The TEY was estimated based on the concentrations of cellulose and Hemi-C, and the TTEY was predicted by multiplying TEY with biomass yield.

Enteric CH$_4$ production from dairy and beef cattle was estimated using nine published models in this study (Table 4). The predictions were based on the relationship between CH$_4$ production (a response variable) and forage nutritive quality (predictor



variables), including the chemical compositions (i.e., CP, NDF, ADF, and ADL) and derived indices (mainly DMI). These models were developed using the dataset from 1) actual measurements of $CH_4$ productions [e.g., respiratory chamber or sulfur hexafluoride (SF6) tracers] and 2) the predictor variables with the lowest root mean square prediction error (RMSPE) shown in the original articles (Ellis et al., 2007; Nielsen et al., 2013). Most of these models required the daily dry matter intake (dDMI, kg d$^{-1}$), which was calculated by multiplying the DMI (g kg$^{-1}$) with the averaged body weight of cows across North America,

Europe, Australia and New Zealand (i.e., 606 kg, from Appuhamy, 2016) in this study. The calculated dDMI was used for further calculations of daily intake (kg d$^{-1}$) of ADF (ADFi) and ADL (ADLi). Some models required the nutritive values that were not analyzed in this study [e.g., the daily metabolizable energy intake (MEi) in model #3 and dietary fatty acid (FA) in models #4 and #5]. In this case, the MEi and FA were considered constants of 162 MJ d$^{-1}$ and 28 g kg$^{-1}$, respectively (Dalley et al., 1999; Ominski et al., 2006; Hegarty et al., 2007).

**2.5 Statistical analysis**

Treatment effects on biomass yield, feedstock chemical compositions, quality indices, and enteric $CH_4$ productions were analyzed using the two-way, repeated-measures Analysis of Variance (ANOVA) using the PROC MIXED procedure in SAS (SAS Institute, Cary, NC, USA). The harvest year (2017-2019), cropping system (forage 2-cut, forage 1-cut, and bioenergy 1-cut), and their interactions were considered fixed factors, while the replicates were considered random. The measurement year

was used as the repeated factor, and each plot was considered as a subject in the repeated measurement. The data normality and homogeneity were assessed by the model-predicted residuals using a Shapiro-Wilk test and equal variance test to meet the ANOVA assumption. Since the feedstock chemical compositions are substantially influenced by harvest management (frequency coupled with timing), the summer- and fall-harvested biomass from the forage 2-cut system were analyzed separately for comparing feedstock fiber compositions, quality indices, tissue nutrients, and $CH_4$ production among other crop

systems [i.e., forage 2-cut (Summer), forage 2-cut (Fall), forage 1-cut, and bioenergy 1-cut]. Also, the first harvest year differed between the two crop systems, so the quality comparisons of the harvested biomass among crop systems only included forage systems in 2017-2019 but contained both forage and bioenergy systems in 2018-2019. Statistical mean differences among treatments were tested using the Tukey method at $\alpha = 0.05$.

**3 Results**

**3.1 General soil and weather information**

The ANOVA test showed that the sampling location and depth significantly affected soil properties even though no location x depth interaction was shown in this study (data not shown). Compared to the west (B1) and central sides (B2) of the buffer strip, the soil in the east (B3) (Table 1) was more acidic and its fertility was lower by showing lower contents of total soil carbon (~18.5% decrease), and other essential nutrients. For instance, the contents of TP, the exchangeable Ca, and Mg were

lower by approximately 7%, 13.8%, and 18.9 %, respectively. The soil EC, highly correlated with the soil nutrient





concentrations (e.g., $NH_4^+$, $K^+$, $Na^+$, $NO_3^-$, $SO_4^{2-}$, and $Cl^-$, etc.), is an alternative indicator for fertility. Although the soil EC (averages across depth) was similar in three sampling locations, the averages across locations showed that the EC in topsoil (0-10 cm) was around 3-4 times higher than deeper soil (<10 cm). Monthly and cumulative precipitation during the study period (2016-2019) and their 30-year average (1990-2019) for this study site are shown in Figure 2. Monthly precipitation

showed an irregular pattern; however, the cumulative data indicated that the precipitation in 2019 was substantially higher than in other years and the 30-year precipitation. During the growing season of perennial grasses (Apr. to Nov.), the cumulative precipitation in 2019 (909 mm) increased by approximately 11% and 18.9%, respectively, compared to the 4-year (820 mm) and 30-year (765 mm) averages. The pattern of monthly temperature during the experimental year followed the 30-average data (Fig. 2c).

**3.2 Biomass yield and nutrient removal**

Two-way interaction between the year (2017-2019) and the cropping system (2-cut and 1-cut forage, and bioenergy crops) for biomass yield showed a significant yield reduction in forage systems but an increase in bioenergy crops (Fig. 3). From 2017 to 2019, biomass yield declined by approximately 30% in the 2-cut forage system (6.3 DM Mg ha$^{-1}$ to 4.4 DM Mg ha$^{-1}$ with a reduction rate of 1.0 Mg ha$^{-1}$ yr$^{-1}$) and 35% in the 1-cut forage system (4.9 DM Mg ha$^{-1}$ to 3.2 DM Mg ha$^{-1}$ with the rate of 0.9

Mg ha$^{-1}$ yr$^{-1}$). By contrast, the biomass yield of bioenergy feedstock increased by 27% from 4.9 DM Mg ha$^{-1}$ (2018) to 6.7 DM Mg ha$^{-1}$ (2019). Based on the harvest frequency, averages across years showed that the 2-cut forage system produced 33% more biomass than the 1-cut forage system (Fig. 3a). The year x cropping system interaction also influenced nutrient removals significantly (Fig. 3b). For both 2- and 1-cut forage, the maximum removal of N, P, K, Ca, and Mg occurred in the first harvest year (2017) and declined over time (Fig. 3b). Nutrient removal is a function of biomass productivity and biomass-nutrient

concentrations, and the decline in nutrient removal corresponded to the decreased biomass yield and nutrient concentrations in the forage system over the three years (Fig. 3 and 4). For instance, the total biomass N removal from the 2-cut forage system reached 106.9 kg ha$^{-1}$ in 2017 and reduced to 52.6 kg ha$^{-1}$ in 2019 (~50.8% reduction); the 1-cut forage system resulted in a total N removal of 84 kg ha$^{-1}$ in 2017, which reduced to 44.5 kg ha$^{-1}$ in 2019 (~47% reduction). For other nutrients, the P, K, Ca, and Mg removals from the 2-cut forage system were 19.8, 132.7, 37.9, and 15.0 kg ha$^{-1}$, respectively, in 2017 and reduced

to 14.8, 93.7, 15.9, and 8.6 kg ha$^{-1}$ in 2019; the 1-cut forage system removed around 18.6 (P), 148.5(K), 25.9(Ca), and 12.4(Mg) kg ha$^{-1}$ in 2017, which reduced to 8.8, 55.7, 15.4, and 6.9 kg ha$^{-1}$ in 2019. Although the overall results of forage systems showed the yearly declined trend for annual yield and nutrient removals (besides P), comparisons between 2018 and 2019 were insignificant. For the bioenergy system, the biomass nutrient removals were also similar in 2018 and 2019 for both primary and secondary nutrients (Fig. 3b).

**3.3 Feedstock chemical compositions and qualities**

Effects of harvest year, cropping system, and their interaction on feedstock chemical compositions, nutrient concentrations, and quality indices were shown in Table 5. Comparisons of these compositional analyses and indices among the forage 2-cut





summer harvest, 2-cut fall harvest, and 1-cut harvest were evaluated in 2017-2019 (forage biomass only), and these comparisons included the bioenergy 1-cut harvest in 2018 and 2019 (forage and bioenergy biomass). For the forage biomass,

both the harvest year (2017-2019) and year x cropping system (i.e., harvest management) interaction significantly influenced the concentrations of NDF, ADF, macronutrients, and feed quality-related indices. Comparisons between forage and bioenergy systems in 2018-2019 showed substantial cropping system effects on all parameters except for Zn. For the forage biomass in 2017-2019, the effects of summer and fall harvest on the NDF and ADF concentrations were similar in the first two years. The NDF and ADF averages across 2017-2018 and three forage harvests were $650.6\pm9.5$ g kg$^{-1}$ and $369.1\pm10.7$ g kg$^{-1}$, respectively

(Table 6). In 2019, the NDF and ADF of the 2-cut forage remained relatively constant under the summer harvest management but significantly decreased by approximately 15% under the fall harvest management, similar to the 1-cut forage. The reduced NDF and ADF concentrations likely enhanced digestibility-related indices (Table 6), and the averages across three forage harvests showed that the 2019 fall-harvested biomass resulted in 12.5, 4.0, 7.3, 6.0, and 26.8% higher DMI, DMD, TDN, NEL, and RFV, respectively than the 2017-2018 averages (Fig. 4b-e). The forage CP concentrations also decreased over the years

with a reduction rate of 12.9 g kg$^{-1}$ yr$^{-1}$ (Fig. 4a). The bioenergy biomass tended to have higher fiber contents than the forage biomass (Table 6). The averages of the bioenergy cellulose, hemicellulose, and lignin concentrations and the predicted TEY in 2018-2019 were 397.9, 299.4, and 66.5 g kg$^{-1}$ of DM and 361.0 L Mg$^{-1}$, respectively, approximately 27.5, 16.7, 43.0, and 22.6% higher than forage feedstock (supplementary figure 1). The TTEY, the product of biomass yield and TEY, from the bioenergy system increased from $1.8\times10^3$ L ha$^{-1}$ in 2018 to $2.4\times10^3$ L ha$^{-1}$ in 2019 (~33% increase shown in Fig. 5). The TTEY

from the forage systems, however, did not show significant differences between 2018 and 2019 even though the overall trend of the forage TTEY was declined. From 2017 to 2019, the TTEY from the forage system declined by 34.2% and 41.9% in the 2-cut and 1-cut forage systems, respectively (Fig. 5).

### 3.4 Nutrient analysis

For nutrient analysis, even though the ANOVA test (Table 5) showed a significant interaction effect between year and cropping

systems on macronutrients, no consistent pattern was observed (supplementary table 1). Averages across three forage harvests showed that most of the macronutrients likely showed the highest concentration in the first harvest year of 2017 (Fig. 4). From 2017 to 2019, the N, K, Ca, and Mg concentrations decreased by approximately 24.1, 17.9, 30.9, and 16.3%, respectively (Fig. 4f). Both forage N and Ca concentrations were below the threshold of the recommended levels for dairy cows, approximately 27.5 g N kg$^{-1}$ DM and 6.2 g Ca kg$^{-1}$ DM shown in the red dashed line (NRC, 2001). For micronutrients, a similar trend of the

declined concentrations was observed for Cu and Zn, reduced by around 22.1 and 10.0%, respectively; by contrast, the concentrations of Mn and Fe increased by 25.9 and 51.0%, respectively, from 2017 to 2019 (Fig. 4g and 4h). Both Cu and Zn were lower than the recommended nutrient levels of 11 and 43 mg kg$^{-1}$ DM. For nutrient ratios, both Ca:P and N:S were reduced substantially by 23.2 and 20.5%, respectively, in the third harvest year, and the N:S ratio was 33% lower than the recommendation (Fig. 4j). In 2018 and 2019, the nutrient concentrations of the bioenergy feedstock were not significantly

different besides Zn. The averages across two years indicated that the nutrient concentrations of the bioenergy feedstock were



generally lower than the forage feedstock by approximately 29-69% and 32-80% for macro- and micro-nutrients, respectively (supplementary table 1).

## 3.5 Methane productions

The enteric $CH_4$ productions were only predicted from forage crops using nine models and the box plots were shown in Figure
6. The averages of the modeled-$CH_4$ ranged from 207.8 to 380.5 g cow$^{-1}$ d$^{-1}$ with the standard deviation (SD) from 4.9 to 31.9 g cow$^{-1}$ d$^{-1}$ (Fig. 6). The M1-M3 models, trained by the dataset of beef cows, resulted in the $CH_4$ average of 234.8±42.1 g cow$^{-1}$ d$^{-1}$ with a coefficient of variation (CV) of 18%. The M4-M6 models, trained by the dataset of dairy cattle, led to a higher $CH_4$ average of 329.1±40.3 g cow$^{-1}$ d$^{-1}$ but a lower variation (CV = 12%) than the M1-M3 predictions. Increases in the M4-M6 average were driven by the highest estimation (380.5±11.2 g cow$^{-1}$ d$^{-1}$) using the M4 model, based on the dDMI, CP, and NDF
predictors, among other models. The increased variation of the M1-M3 predictions, on the other hand, was due to the highest variation (CV = 17%) from the M1 model, based on the dDMI and ADLi predictors. The $CH_4$ predictions using the M7-M9 models, built based on the integrated dataset of beef and dairy cows, resulted in an average of 231.9±19.1 g cow$^{-1}$ d$^{-1}$ with the lowest variation (CV = 8%). For the individual model, the two-way ANOVA showed that most of the $CH_4$ predictions were sensitive to year variation, but only the M5-, M7-, and M8-predicted $CH_4$ significantly responded to the harvest management
and their interaction (supplementary table 2). The averages from each category (e.g., M1-M3 average from the beef category) showed that the model-$CH_4$ increased over the years with the rates of 11.0 (beef), 8.1 (dairy), and 8.4 (combined) g cow$^{-1}$ d$^{-1}$, respectively (Table 7). No significant year x forage system interaction was observed for the $CH_4$ predicted from beef and dairy categories. The interaction effect was only shown in the "combined" category (*p*-value = 0.0074). In the "combined" category, the modeled-$CH_4$ remained stable (i.e., 221.1~228.7 g cow$^{-1}$ yr$^{-1}$) in the first two harvest years of 2017 and 2018 but increased
by approximately 10.8% under the fall harvest management in 2019 (i.e., 250.9 g cow$^{-1}$ d$^{-1}$) compared to the 2017-2018 average (i.e., 226.4 g cow$^{-1}$ d$^{-1}$).

## 4 Discussion

### 4.1 Biomass yield and nutrient removal

This study evaluated the feasibility of utilizing forage (perennial cool-season grass mixtures) and bioenergy (warm-season
grass mixtures) systems for establishing a sustainable buffer strip (Table 2). For the forage buffer, Kelly et al. (2007) reported that the monoculture SB and AL produced annual biomass yields ranging from 5-6 Mg ha$^{-1}$ and 5-7 Mg ha$^{-1}$, respectively. The polyculture forage systems (PR-colver or SB-TG-Kentucky bluegrass mixtures) established in riparian areas produced annual yields from 2 to 10 Mg ha$^{-1}$, and the yields of grass mixtures cultivated on other marginal lands (e.g., on the CRP registered lands) were from 2.8 Mg ha$^{-1}$ (the AL-pubescent wheatgrass mixture in Montana), 3.4 (the TF-OR mixture in Georgia), 4.2
(the red clover-TF mixture in Missouri), respectively (Anderson er al., 2016; Christen et al., 2013; Lee et al., 2013; Tufekcioglu et al., 2003). For the bioenergy buffer, the monoculture SW showed a range of annual yield from 4 to 13 Mg ha$^{-1}$, and the grass



mixtures (mainly SW, PC, BB, and IN) ranged from 2.8 to 10.7 Mg ha$^{-1}$ (Cooney et al., 2023; Ferrarini et al., 2017; Gamble et al., 2016; Gopalakrishnan et al., 2012; Kelly et al., 2007; Tufekcioglu et al., 2003; Zamora et al., 2013). This yield variation was likely due to the confounding effects of species, growth environment, and management practices (Cooney et al., 2023).
Thus, the perennial buffers in this study showed their economic potential by producing decent biomass yield (forage: 3.2-6.3 Mg ha$^{-1}$; bioenergy yield: 4.9-6.7 Mg ha$^{-1}$), especially under no fertilizer application.

Compared to the bioenergy system, greater nutrient concentrations in forage biomass (supplementary table 1) indicated that forage-type grasses have higher nutrient demand (Kering et al., 2012; Pedroso et al., 2014). The increased nutrient requirement
also referred that cool-season forages require more available soil nutrients (e.g., high N fertilization) to produce similar amounts of dry biomass as warm-season bioenergy grasses, also suggesting that more nutrients are likely removed from soils by cool-season forage grasses (Minson, 1981; Mullahey et al., 1992; Follett and Wilkinson, 1995; Kering et al., 2012; Pedroso et al., 2014). In this study, the higher nutrient removal of the cool-season grasses resulted in possible depletion of soil nutrients over time (even though the yearly changes in soil fertility were not assessed), presumably accounting for a gradual decline in
biomass yield in the forage system (approximately 0.9 Mg ha$^{-1}$ yr$^{-1}$ reduction). From 2017 to 2019, decreases in biomass-N, -K, -Ca, and -Mg concentrations inferred that soil nutrients, including the nutrient transported from the main C-S field via runoff, might be inadequate to fully support the growth of the cool-season grasses in the forage system. Harvest management (frequency and timing) also influences biomass yield and crop nutrient removal. Our results support previous findings in which total annual biomass yield and nutrient removal are usually increased by multiple cuttings in the forage system (Schultz et al.,
1995; Fike et al., 2006b; Mitchell and Schmer, 2012). Harvest timing also plays a critical role in stand persistence and the 'regrowth vigor' potential, which influences long-term buffer strip production and sustainability (Mulkey et al., 2006; Mitchell et al., 2014). In this study, the delayed harvest to the end of the growing season was implemented in both forage systems or until after a killing frost for the bioenergy system to improve feedstock stand longevity by providing extended time for vegetative development and reproductive tiller growth and to translocate nutrients to underground crop tissues that can be
recycled for use in the following year (MacAdam and Nelson, 2003; Lee et al., 2014; Zumpf et al., 2019).

Compared to the forage systems in which biomass yield and nutrient concentrations declined over time, the bioenergy system showed an increased yield potential under continuously unfertilized conditions (Fig. 3). There was no biomass harvest in the bioenergy system in 2017 due to the reestablishment that year. Therefore, the first harvest year for the bioenergy biomass was
in 2018. Based on the harvest year, the side-by-side comparisons showed that the 2-cut forage system produced a higher amount of biomass than the bioenergy system in the first harvest year, but its biomass yield declined in the following years; conversely, the bioenergy system showed the highest yield potential in the second harvest year for any system (Fig. 3a). Although the bioenergy system was only harvested for two successive years in our case, many studies report that warm-season grasses have high nutrient-use efficiency and consistent biomass production across years (Brown, 1978; Sage et al., 1987;
Ghannoum et al., 2011; Sage and Zhu, 2011; van der Weijde et al., 2013; Pedroso et al., 2014; Lee et al., 2018). For example,





Lee et al. (2018) showed that maximum bioenergy feedstock biomass yield occurred in the third year after the establishment (~2 times more than in the establishment year) and the stable yield production can be continuous for up to 7 years. For scavenging nutrients moved down from the C-S field, however, a cool-season grass-based forage system can be considered an ideal short-term candidate for riparian zone filter strips by showing more effective erosion control, sediment trapping, and

nutrient removal than a warm-season grass-based bioenergy system (Lynn, 2004). Based on the first two harvest events (2017 and 2018 in the forage system and 2018 and 2019 in the bioenergy system), the 2-cut and 1-cut forage systems resulted in total N removal of 176.2 kg ha⁻¹ and 128.0 kg ha⁻¹ respectively in the harvested biomass, while only 60.3 kg ha⁻¹ was removed from the bioenergy system (Fig. 3b). These substantial N removal from the forage system mainly resulted from significantly high yield and biomass nutrient concentrations in the first harvest year (2017). On the other hand, the bioenergy system can be

considered for long-term nutrient loss reduction plans as it produced consistently high biomass.

## 4.2 Feedstock quality

Biomass quality characteristics are defined by its use as livestock feed or biofuel feedstock. As forage feedstocks, biomass CP and crude fibers (NDF and ADF) are critical factors for animal performance and quality, such as livestock weight or milk production (Assefa and Ledin, 2001; Collins and Fritz, 2003). The biomass NDF and ADF can be used to evaluate the

digestibility, palatability, and energy level of animal feeds based on their predictions of DMD, DMI, TDN, NEL, and RFV indices (Guretzky et al., 2011). For example, the higher NDF generally lowers the ingestion of dry matter (DMI), and the higher ADF likely reduces the overall digestibility (DMD), digestible nutrients (TDN), and energy level (NEL) of forage feeds for animals and lowers forage qualities (Collins and Fritz, 2003). Delayed harvest usually reduced biomass CP and increased fiber concentrations resulting from the N translocation to belowground rhizomes; however, no significant harvest effects on

CP or crude fibers (NDF and ADF) were shown in this study (Table 6), presumably due to a substantial environmental/climatic impact. Without direct N fertilizer replenishment, the three years of continuous crop harvests resulted in decreases in both biomass yield and CP content in the forage systems (Fig. 3 and 4a). The forage NDF also gradually reduced with increasing N depletion over the years (Fig.4) even though several studies showed that the cell wall compositions were not influenced by different N input/soil N contents (Liu et al., 2015; Ameen et al., 2019). Several abiotic (e.g., severe drought, salinity, heat) or

biotic (diseases or pests) stresses might damage cell structure, inhibit crop growth, stunt tissue development, and reduce structural cell wall compositions (Hoover et al., 2018; Fan et al., 2020). Thus, the abrupt NDF reduction in (~10.8%) in 2019 presumably resulted from the harsh environmental stresses (i.e., the severe leaching and nutrient depletion resulting from intense precipitation in 2019, Fig. 2).

Mineral levels in forage are also essential for both livestock health and performance and the effectiveness of ruminal microorganisms for fiber digestion. The mineral concentrations in biomass usually declined with advancing maturity (Fleming, 1973); however, this study did not observe the reduced mineral content in forage feedstock between two harvest timing (at the anthesis in summer [June] vs. after complete senescence in fall [October/November]) possibly resulting from a substantial



environmental effect. Besides the N content, Ca and P deficiency are often considered critical among other macronutrients
because of their roles in the development of skeletal structure, metabolism, and milk production. Under the continuously
unfertilized management, the forage P concentrations (average across forage harvests) remained fairly adequate for cattle
requirements (~3.2 g kg$^{-1}$). The continuously declining Ca concentrations, however, showed the feed Ca content was
insufficient for a dairy cow's daily diet (minimum requirement ~ 6.2 g kg$^{-1}$ DM; NRC, 2001). A long-term Ca deficiency in
animal diet could result in abnormalities of bond development and depression of milk yield (McDowell, 1992; Suttle, 2022).
Potassium, as the primary intracellular cation, recommendation for dairy cows ranges from 6 to 12 g kg$^{-1}$ of DM and its
deficiency is rarely observed in forage biomass given the high K content in grasses and legumes (Marijanušić et al., 2017).
Although excess dietary K uptake (e.g., 20-24 g kg$^{-1}$ DM in this study) could interfere with Ca homeostasis, K toxicity is rare
in cattle because dairy cattle showed a great ability to easily excrete excessive K intake (NRC, 2005). Most of the
micronutrients act as critical components of metalloenzymes or metalloproteins, significantly related to the metabolic function,
immune system, and antioxidant status of the livestock (Suttle, 2022). The Cu, Zn, and Fe deficiency in ruminant grazing
forages is a widespread issue in many areas of the world and is often required for supplementation (McDowell, 1992; Spears,
1994; Marijanušić et al., 2017). This study also showed the Cu and Zn deficiency in forages but not for Fe. Both Cu and Zn
concentrations were below the critical levels by approximately 50 and 43%, respectively. Furthermore, appropriate nutrient
ratios are as critical as individual elements for animal growth and performance. For instance, the Ca:P ratio is important for
supporting appropriate bone development (especially for young growing animals) and lactating cows for milk production. The
ideal Ca:P ratio was recommended from 1.5:1 to 2.0:1 for dairy cows, but the ratio of the third-year forage (average across
forage systems) was below this threshold due to substantial decreases in the tissue Ca concentration (Fig. 4i). The N:S ratio is
another important factor for ruminant microbial protein synthesis and S-contain amino acid production (NRC, 2005). The N:S
ratio in this study (5.7~7.3) was way below the recommended ratio ranging from 10:1 to 12:1.


As bioenergy feedstocks, the increased cell wall contents often observed in warm-season grasses are generally considered
indications of desirable biofuel quality, especially the structural carbohydrates of cellulose and hemicellulose for producing
bioethanol via the biochemical (i.e., fermentation) process (Li et al., 2016). Previous studies (Hong et al., 2012; Guo et al.,
2017; Ameen et al., 2019) showed that the cellulose concentrations of the big bluestem (BB), Indiangrass (IND), switchgrass
(SW), and prairie cordgrass (PC) monocultures were in ranges of 378-420, 380-451, 360-401, and 400-421 g kg$^{-1}$, respectively.
The polyculture system of BB, IND, and SW mixtures showed a similar cellulose range from 360 to 425 g kg$^{-1}$. For
hemicellulose, the concentration ranges were 308-310 (BB), 278-315 (IND), 310-325 (SW), and 293-312 (PC) g kg$^{-1}$,
respectively, and grass mixtures were also under a similar range (300-315 g kg$^{-1}$). This study showed that the grass-mixture
bioenergy buffer can also offer feedstock with reasonable qualities (i.e., a two-year average of cellulose~ 397.9 g kg-1;
hemicellulose~299.4 g kg$^{-1}$) and a great potential for increasing ethanol yield productions based on the increased TTEY from
$1.8 \times 10^3$ (2018) to $2.4 \times 10^3$ L ha$^{-1}$ (2019).





### 4.3 Environmental impact

The forage nutritive values not only influence cattle performance but highly correlate to the $CH_4$ production in the rumen. For
instance, the increased feed quality likely increases feed consumption, usually expressed by DMI, and the increased DMI further accelerates cattle $CH_4$ emissions. Instead of direct $CH_4$ measurements, several empirical models have been used for the $CH_4$ predictions based on DMI, which is the most often used predictor, and other attributes, such as CP, NDF, ADF, and ADL (Ellis et al., 2007; Storlien et al., 2014; Appuhamy et al., 2016). Although the DMI of individual cows was not routinely measured due to the limited budget in commercial farms, the estimated DMI can also predict $CH_4$ emissions with a performance
as good as the measured DMI using suitable models (Appuhamy et al., 2016). Enteric $CH_4$ emissions are highly variable among ruminants. Generally speaking, the species with heavier body weight tend to consume more feed as well as produce more $CH_4$ via feed fermentation (e.g., cattle > goat or sheep). For example, the average body weight of beef cattle was approximately 450 kg per cow with a daily DMI (dDMI) of 8.0 kg in North America, and the $CH_4$ emissions ranged from 50 to 250 g cow$^{-1}$ d$^{-1}$. The average weight of dairy cattle was around 644 kg with a dDMI of 21 kg, producing $CH_4$ from 200 to 600 g cow$^{-1}$ d$^{-1}$
(Ellis et al., 2007; Appuhamy et al., 2016; Hales et al., 2022). This study also showed a similar range of the $CH_4$ estimations, and the predicted $CH_4$ emissions increased with advancing DMI in the third harvest year. Furthermore, it is possibly misguiding for farmers to optimize livestock management practices based on a simple judgment of the overall $CH_4$ emissions. For instance, although the overall $CH_4$ emissions of the dairy cows were higher (128 kg cow$^{-1}$ yr$^{-1}$) in North America than in the European Union (117 kg cow$^{-1}$ yr$^{-1}$) and Oceanian (99 kg cow$^{-1}$ yr$^{-1}$) cows, the higher milk yield led to lower emission intensity in North
America (FAO, 2014). Since livestock production is a consequence of the overall feed quality, which needs to consider both digestibility (e.g., DMD) and nutrient levels (e.g., CP and minerals), the overall high-quality forage likely increases animal production as well as mitigates $CH_4$ emission intensity (Lee et al., 2017).

### 5 Conclusions

Perennial grass mixtures are ideal polyculture systems for building productive and sustainable buffer stripes. The cool-season
forage and warm-season bioenergy grasses showed different strengths as buffers by serving specific purposes. The forage-type buffer can be an ideal short-term candidate for riparian areas with high leaching potential due to its great efficiency of nutrient scavenging, which can be further improved under the multiple harvest management. The high nutrient demand of forage crops, however, likely compromised buffer sustainability under the successive nutrient starvation condition. From a quality perspective, the successively-harvested forage buffer without nutrient input through fertilizer application was incapable of
providing livestock with adequate nutritive values, especially yearly reductions of crucial protein and major mineral contents (i.e., Ca, Cu, Zn), even though the forage digestibility seemed to increase in the third harvest year. The overall low-quality feed likely lower ruminants' performance and possibly aggravates the impact of enteric $CH_4$ emissions on the global greenhouse gas burden. On the other hand, although this is a short-term study, the bioenergy-type buffer showed better sustainability than the forage buffer and a potential of continuous and stable yield supply based on our previous and other long-



term studies, which could provide local stakeholders with a long-term opportunity of offering extra economic benefits and ecosystem services, simultaneously.

*Data Accessibility.* Data can be directly obtained by contacting the lead author.

*Supporting Information.* Additional information may be found online in the supporting information tab for this article.

*Competing interests.* This research was funded by the Metropolitan Water Reclamation District (MWRD) of Greater Chicago. Any opinions, findings, and recommendations expressed in this paper are those of the author(s) and do not necessarily reflect the views of the MWRD of Greater Chicago. The authors do not have any other relevant affiliations or financial involvement
with any organization with a financial interest in or financial conflict with the subject matter or materials discussed in the manuscript apart from those disclosed.

*Acknowledgements.* This work was supported by the MWRD of Greater Chicago as part of the collaboration, the crop and field management from Jacob Baylor, and technical assistance from the Soils laboratory of MWRD.



**Table 1. Basic physical and chemical properties of soil profiles (0-100 cm) of the buffer strips prior to the study in 2016. Three blocks were compared from different depths using a two-way ANOVA with a significance level of 0.05. The block (B), soil depth, and their interactions were considered fixed factors, while the replicates were considered random. The lowercase letters indicate mean separation organized from highest to lowest value for each column using the Tukey test (no mean separations were applied if the variable effect was not significant).**

| | Factors | BD (g cm$^{-3}$) | pH | EC (dm m$^{-1}$) | *TC | *TN | TP | K | Ca | Mg |
|---|---|---|---|---|---|---|---|---|---|---|
| | | | | | -------------- (g kg$^{-1}$) -------------- | | | --- †Exchangeable (mg kg$^{-1}$) --- | | |
| Location | West (B1) | 1.67 | 6.91a | 0.20 | 16.1a | 1.13 | 0.56a | 107.1 | 2391.7a | 500.5a |
| | Central (B2) | 1.67 | 7.05a | 0.23 | 15.3ab | 1.11 | 0.54ab | 112.7 | 2477.8a | 507.9a |
| | East (B3) | 1.69 | 6.36b | 0.23 | 12.8b | 1.05 | 0.51b | 105.6 | 2098.1b | 409.1b |
| Soil depth (cm) | 0-5 | 1.35c | 6.59c | 0.52a | 17.7a | 1.49a | 0.65a | 176.4a | 2551.4a | 460.4 |
| | 5-10 | 1.55b | 6.52c | 0.33b | 17.8a | 1.48a | 0.62a | 154.8a | 2344.1ab | 442.7 |
| | 10-20 | 1.80a | 6.35c | 0.16c | 14.5ab | 1.15b | 0.53b | 102.7b | 2203.2ab | 488.1 |
| | 20-30 | 1.83a | 6.64c | 0.11c | 12.9b | 0.93c | 0.48bc | 79.9bc | 2040.4b | 489.0 |
| | 30-60 | 1.77a | 7.04b | 0.10c | 11.7b | 0.73d | 0.45c | 70.2bc | 2316.1ab | 485.7 |
| | 60-100 | 1.76a | 7.48a | 0.10c | 13.8ab | 0.78cd | 0.51bc | 66.8c | 2480.1a | 469.2 |

BD: bulk density; TC: total carbon; TN: total nitrogen; TP: total phosphorus (direct colorimetric method); *: TC and TN were determined by dry combustion method; †: exchangeable nutrients (K, Ca, Mg) determined by the Mehlich-3 method.






**Table 2 Cropping systems, plant species composition, harvest frequency (once and twice), and harvest dates from 2017 to 2019 for perennial buffer strips established at Fulton County, IL.**

| Cropping system (CS) | Crop types and species | Harvest frequency (timing) | Harvest date 2017 | 2018 | 2019 |
|---|---|---|---|---|---|
| F (2-cut; Sum.) | Forage (cool-season mixtures, including AL, MF, OR, PR, SB, TF, TG) | Twice (Summer & Fall) | Jun. 26 | Jun. 05 | Jun. 26 |
| F (2-cut; Fall) | | | Nov. 01 | Oct. 30 | Nov. 06 |
| F (1-cut; Fall) | | Once (Fall) | Nov. 01 | Oct. 30 | Nov. 06 |
| †B (1-cut; Fall) | Bioenergy (warm-season mixtures, including BB, IN, PC, SW) | Once (Fall) | N/A | Oct. 30 | Nov. 06 |

F (2-cut; Sum.): forages with two-harvest management (1st harvest in Summer ranging from Jun. 1st to Aug. 31st)

F (2-cut; Fall): forages with two-harvest management (2nd harvest in Fall ranging from Sept. 1st to Nov. 31st)

F (1-cut; Fall): Forages with one-harvest management in Fall

B (1-cut; Fall): Bioenergy crop with one harvest management in Fall

†Bioenergy crops were not harvested in 2017 for biomass nutrient analysis due to insufficient biomass production, and this system was reestablished in 2017.

AL: alfalfa; BB: big bluestem; IN: Indiangrass; MF: meadow fescue; OR: orchardgrass; PC: prairie cordgrass; PR: perennial ryegrass;

SB: smooth bromegrass; SW: switchgrass; TF: tall fescue; TG: Timothy grass.



**Table 3. Calculations of feedstock quality indices.**

| Indices (abbreviation, unit) | Equations | Descriptions |
|---|---|---|
| Crude protein (CP, g kg$^{-1}$) | $CP = N \times 6.25$ | • An amount of protein in a feed, estimated from the measured N content |
| Dry matter intake (DMI, g kg$^{-1}$) | $DMI = 120 / NDF$ | • The measured amount of feed an animal consumes per day on a dry basis |
| Dry matter digestibility (DMD, g kg$^{-1}$) | $DMD = 88.9 - (0.779 \times ADF)$ | • The portion of the dry matter in a feed that is digested by livestock at a certain level of feed intake |
| Total digestible nutrients (TDN, g kg$^{-1}$) | $TDN = (-1.291 \times ADF) + 101.35$ | • The sum of the CP, lipid, digestible fiber, and non-structural carbohydrates |
| Net energy for lactation (NEL, g kg$^{-1}$) | $NEL = [1.044 - (0.0119 \times ADF)] \times 2.205$ | • An estimate of energy in a feed that is available for body maintenance and milk production |
| Relative feed value (RFV, %) | $RFV = DMD \times DMI \times 0.775$ | • An estimate of value combining a forage's digestibility and intake potential |
| Cellulose (g kg$^{-1}$) | $Cellulose = ADF - ADL$ | • A structural carbohydrate mainly made up of long-chain polymers of glucose |
| Hemicellulose (Hemi-C, g kg$^{-1}$) | $Hemi\text{-}C = NDF - ADF$ | • A structural carbohydrate mainly made up of long-chain polymers of xylose |
| [†]Theoretical ethanol yield (TEY, L Mg$^{-1}$) | $TEY = (Cellulose + Hemi\text{-}C) \times F1 \times F2 \times F3 \times F4 \times 1000/\rho$ | • An estimate of ethanol yield derived from cellulose and hemicellulose contents using biological conversion techniques |
| Total theoretical ethanol yield (TTEY, L ha$^{-1}$) | $TTEY = TEY \times$ dry biomass yield | • An estimate of total TEY based on the product of TEY and biomass yield |

[†]: $F1$ = 0.51 (the coefficient of conversion of sugar to ethanol); $F2$ = 0.85 (the conversion efficiency of sugar to ethanol; $F3$ = 1.11 (the coefficient of conversion of cellulose and hemicellulose to sugar; $F4$ = 0.85 (the conversion efficiency of cellulose and hemicellulose to sugar); $\rho$ = 0.79 g ml$^{-1}$ (the specific gravity of ethanol)






**Table 4. A summary of the published equations for predicting methane (CH₄) production based on feedstock nutritive quality.**

| Model | Equations for CH₄ estimation (MJ d⁻¹)[a] | Cattle[b] | RMSPE[c] | $R^2$ | Refs |
|---|---|---|---|---|---|
| 1 | $= 2.30 + 1.12 \times$ dDMI (kg d⁻¹) $- 6.26 \times$ ADLi (kg d⁻¹) | Beef | 18.7 | 0.74 | Ellis et al., 2007 |
| 2 | $= 5.70 + 1.41 \times$ ADFi (kg d⁻¹) | Beef | 24.3 | 0.56 | Ellis et al., 2007 |
| 3 | $= 2.94 + 0.0585 \times$ MEi (MJ d⁻¹) $+ 1.44 \times$ ADFi (kg d⁻¹) $- 4.16 \times$ ADLi (kg d⁻¹) | Beef | 14.4 | 0.85 | Ellis et al., 2007 |
| 4 | $= 1.36 \times$ dDMI (kg d⁻¹) $- 0.125 \times$ FA (g kg⁻¹) $- 0.02 \times$ CP (g kg⁻¹) $+ 0.017 \times$ NDF (g kg⁻¹) | Dairy | 13.8 | 0.77 | Nielsen et al., 2013 |
| 5 | $= 1.23 \times$ dDMI (kg d⁻¹) $- 0.145 \times$ FA (g kg⁻¹) $+ 0.012 \times$ NDF (g kg⁻¹) | Dairy | 14.3 | 0.75 | Nielsen et al., 2013 |
| 6 | $= 5.87 + 2.43 \times$ ADFi (kg d⁻¹) | Dairy | 35.4 | 0.56 | Ellis et al., 2007 |
| 7 | $= 3.272 + 0.736 \times$ dDMI (kg d⁻¹) | Combined | 28.2 | 0.68 | Ellis et al., 2007 |
| 8 | $= 3.41 + 0.520 \times$ dDMI (kg d⁻¹) $- 0.996 \times$ ADFi (kg d⁻¹) $+ 1.15 \times$ NDFi (kg d⁻¹) | Combined | 30.5 | 0.67 | Ellis et al., 2007 |
| 9 | $= 3.69 + 0.543 \times$ dDMI (kg d⁻¹) $+ 0.698 \times$ NDFi (kg d⁻¹) $- 3.26 \times$ ADLi (kg d⁻¹) | Combined | 29.6 | 0.71 | Ellis et al., 2007 |

[a]: dDMI (kg d⁻¹): daily dry matter intake, calculated by multiplying DMI (g kg⁻¹) and the averaged body weight of cow (606 kg, from Appuhamy, 2016); ADFi (kg d⁻¹): daily acid detergent fiber intake; ADLi (kg d⁻¹): daily acid detergent lignin intake; MEi (MJ d⁻¹): daily metabolizable energy intake (~162); FA (g kg⁻¹): dietary fatty acid (~ 28); CP (g kg⁻¹): crude protein concentration of DM; NDF: neutral detergent fiber concentration of DM.

[b]. Combine = beef + dairy.

[c]: RMSPE: square root of mean square prediction error reported by the original papers.





**Table 5. Analysis of variance (ANOVA) showed the effects of main factors, including year (Y), cropping system (CS), and interactions on biomass fiber analysis, quality indices, and nutrients analysis of the forage and bioenergy feedstocks of the buffer strips with a significance level of 0.05.**

| Parameters | 2017-2019 (forage only) | | | 2018-2019 (forage + bioenergy) | | |
|---|---|---|---|---|---|---|
| | Y | CS | Y x CS | Y | CS | Y x CS |
| ---------------------------- Fiber analysis ---------------------------- | | | | | | |
| NDF (g kg$^{-1}$) | **** | * | ** | *** | *** | * |
| ADF (g kg$^{-1}$) | ** | ns | * | ** | ** | * |
| ADL (g kg$^{-1}$) | ns | ns | ns | ns | * | ns |
| ---------------------------- Quality indices ---------------------------- | | | | | | |
| DMI (g kg$^{-1}$) | **** | * | ** | *** | *** | ** |
| DMD (g kg$^{-1}$) | ** | ns | * | ** | ** | * |
| TDN (g kg$^{-1}$) | ** | ns | * | ** | ** | * |
| NEL (g kg$^{-1}$) | ** | ns | * | ** | ** | * |
| RFV (%) | **** | * | ** | *** | ** | ** |
| Cellulose (g kg$^{-1}$) | ** | * | ns | ** | ** | ns |
| Hemi-C (g kg$^{-1}$) | **** | ns | ns | ** | *** | ns |
| TEY (L Mg$^{-1}$) | **** | ** | ns | *** | *** | ns |
| Ash (g kg$^{-1}$) | **** | * | ns | **** | ** | ns |
| ---------------------------- Nutrients analysis ---------------------------- | | | | | | |
| N/CP (g kg$^{-1}$) | **** | ns | *** | * | *** | ** |
| P (g kg$^{-1}$) | *** | ns | **** | ns | **** | ** |
| K (g kg$^{-1}$) | ** | ns | *** | ** | **** | * |
| S (g kg$^{-1}$) | ** | ns | * | ns | **** | * |
| Mg (g kg$^{-1}$) | **** | ns | * | ns | **** | ** |
| Ca (g kg$^{-1}$) | **** | ns | * | * | **** | ns |
| B (mg kg$^{-1}$) | *** | ns | ns | ** | * | ns |
| Zn (mg kg$^{-1}$) | * | ns | ns | * | ns | ** |
| Mn (mg kg$^{-1}$) | * | * | ns | * | *** | ns |
| Fe (mg kg$^{-1}$) | *** | ns | ** | ** | * | * |
| Cu (mg kg$^{-1}$) | ** | ns | ns | ns | * | ns |
| Al (mg kg$^{-1}$) | *** | ns | ** | ** | * | ** |
| Ca:P | *** | ns | *** | * | * | * |

Level-1 (*): $0.05 < p < 0.01$; Level-2 (**): $0.01 < p < 0.001$; Level-3 (***): $0.001 < p < 0.0001$;

Level-4 (****):$p < 0.0001$; ns: not significant


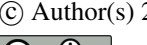



**Table 6.** Fiber analysis and the derived forage quality indices of the harvested biomass, influenced by the year and cropping system interaction (Y x CS) from 2017 to 2019. The superscript lowercase letters indicate the mean separation ($\alpha$=0.05) of the forage crops collected from 2017–2019. The subscript uppercase letters indicate the mean separation of both forage and bioenergy crops collected from 2018 and 2019 (no mean separations were applied if the variable effect was not significant).

| Year | Cropping system | NDF | ADF | ADL | DMI | DMD | TDN | NEL | CP | RFV (%) |
|---|---|---|---|---|---|---|---|---|---|---|
| | | ------ Fiber analysis (g kg$^{-1}$) ------ | | | ------ Quality indices (g kg$^{-1}$) ------ | | | | | |
| 2017 | F (2-cut; Sum.) | 655.6$^a$ | 361.8$^{ab}$ | 49.3 | 18.3$^b$ | 607.2$^{bc}$ | 546.4$^{bc}$ | 1.4$^{ab}$ | 108.8$^a$ | 86.1$^b$ |
| | F (2-cut; Fall) | 660.2$^a$ | 363.9$^{ab}$ | 76.0 | 18.2$^b$ | 605.5$^{bc}$ | 543.6$^{bc}$ | 1.3$^{ab}$ | 105.0$^{ab}$ | 85.3$^b$ |
| | F (1-cut; Fall) | 641.9$^a$ | 354.0$^{abc}$ | 55.1 | 18.7$^b$ | 613.3$^{abc}$ | 556.5$^{abc}$ | 1.4$^{ab}$ | 106.3$^{ab}$ | 89.0$^b$ |
| 2018 | F (2-cut; Sum.) | 653.3$^a_B$ | 379.5$^{ab}_{BCD}$ | 48.5 | 18.4$^b_{BC}$ | 593.4$^{bc}_{ABC}$ | 523.6$^{bc}_{ABC}$ | 1.3$^b_{ABC}$ | 104.4$^{ab}_A$ | 84.5$^b_{CD}$ |
| | F (2-cut; Fall) | 636.0$^a_{BC}$ | 376.1$^{ab}_{BC}$ | 53.4 | 18.9$^b_B$ | 596.0$^{bc}_{BC}$ | 527.9$^{bc}_{BC}$ | 1.3$^b_{BC}$ | 81.9$^{bcd}_{AB}$ | 87.6$^b_{BC}$ |
| | F (1-cut; Fall) | 656.4$^a_B$ | 379.3$^{ab}_{BCD}$ | 48.6 | 18.3$^b_{BC}$ | 593.5$^{bc}_{ABC}$ | 523.8$^{bc}_{ABC}$ | 1.3$^b_{ABC}$ | 78.1$^{cd}_{AB}$ | 84.1$^b_{CD}$ |
| | †B (1-cut; Fall) | 775.3$_A$ | 468.7$_A$ | 66.0 | 15.5$_D$ | 523.9$_D$ | 408.4$_D$ | 1.1$_D$ | 42.5$_{CD}$ | 62.9$_E$ |
| 2019 | F (2-cut; Sum.) | 642.9$^a_B$ | 386.7$^a_{ABCD}$ | 57.4 | 18.7$^b_B$ | 587.7$^c_{ABCD}$ | 514.2$^c_{ABCD}$ | 1.3$^b_{ABCD}$ | 65.0$^d_{BC}$ | 85.2$^b_{CD}$ |
| | F (2-cut; Fall) | 547.8$^b_D$ | 302.9$_D$ | 37.2 | 22.0$^a_A$ | 653.1$^a_A$ | 622.5$^a_A$ | 1.5$^a_A$ | 92.5$^{abc}_{AB}$ | 111.2$^a_A$ |
| | F (1-cut; Fall) | 554.1$^b_{CD}$ | 326.6$^{bc}_{CD}$ | 34.1 | 21.7$^a_A$ | 634.6$^{ab}_{AB}$ | 591.8$^{ab}_{AB}$ | 1.4$^{ab}_{AB}$ | 86.3$^{abcd}_{AB}$ | 106.8$^a_{AB}$ |
| | †B (1-cut; Fall) | 752.3$_A$ | 460.1$_{AB}$ | 67.1 | 16.0$_{CD}$ | 530.6$_{CD}$ | 419.5$_{CD}$ | 1.1$_{CD}$ | 23.8$_D$ | 65.8$_{DE}$ |

F (2-cut; Sum.): Forage crop with two harvest management (1st harvest in Summer)

F (2-cut; Fall): Forage crop with two harvest management (2nd harvest in Fall)

F (1-cut; Fall): Forage crop with one harvest management in Fall

B (1-cut; Fall): Bioenergy crop with one harvest management in Fall

†Bioenergy crops were not harvested in 2017 for biomass nutrient analysis due to insufficient biomass production.



**Table 7. The model-predicted CH₄ productions of the forage feedstock, influenced by the year and cropping system interaction (Y x CS) from 2017 to 2019. The superscript lowercase letters indicate the mean separation (α=0.05) of the forage crops collected from 2017-2019 (no mean separations were applied if the variable effect was not significant).**

| Factor | | Beef | Dairy | Combined |
|---|---|---|---|---|
| Y | CS | ----------- $CH_4$ (g cow$^{-1}$ d$^{-1}$) ----------- | | |
| 2017 | F (2-cut; Sum.) | 231.1 | 321.1 | 227.4$^b$ |
| | F (2-cut; Fall) | 212.5 | 321.2 | 221.1$^b$ |
| | F (1-cut; Fall) | 227.7 | 322.7 | 228.6$^b$ |
| 2018 | F (2-cut; Sum.) | 235.3 | 324.9 | 226.7$^b$ |
| | F (2-cut; Fall) | 234.8 | 330.8 | 228.7$^b$ |
| | F (1-cut; Fall) | 234.7 | 327.5 | 226.4$^b$ |
| 2019 | F (2-cut; Sum.) | 232.8 | 333.1 | 225.9$^b$ |
| | F (2-cut; Fall) | 248.8 | 338.7 | 252.1$^a$ |
| | F (1-cut; Fall) | 255.6 | 342.0 | 249.8$^a$ |
| | *P-value* | *ns* | *ns* | ** |
| Y mean | 2017 | 223.8$^b$ | 321.7$^c$ | 225.7$^b$ |
| | 2018 | 235.0$^{ab}$ | 327.8$^b$ | 227.3$^b$ |
| | 2019 | 245.7$^a$ | 337.9$^a$ | 242.6$^a$ |
| | Slope (R$^2$) | 11.0 (1.0) | 8.1 (1.0) | 8.4 (0.8) |
| | *p-value* | ** | **** | *** |

Level-1 (*): $0.05 < p < 0.01$; Level-2 (**): $0.01 < p < 0.001$; Level-3 (***): $0.001 < p < 0.0001$; Level-4 (****): $p < 0.0001$; ns: not significant






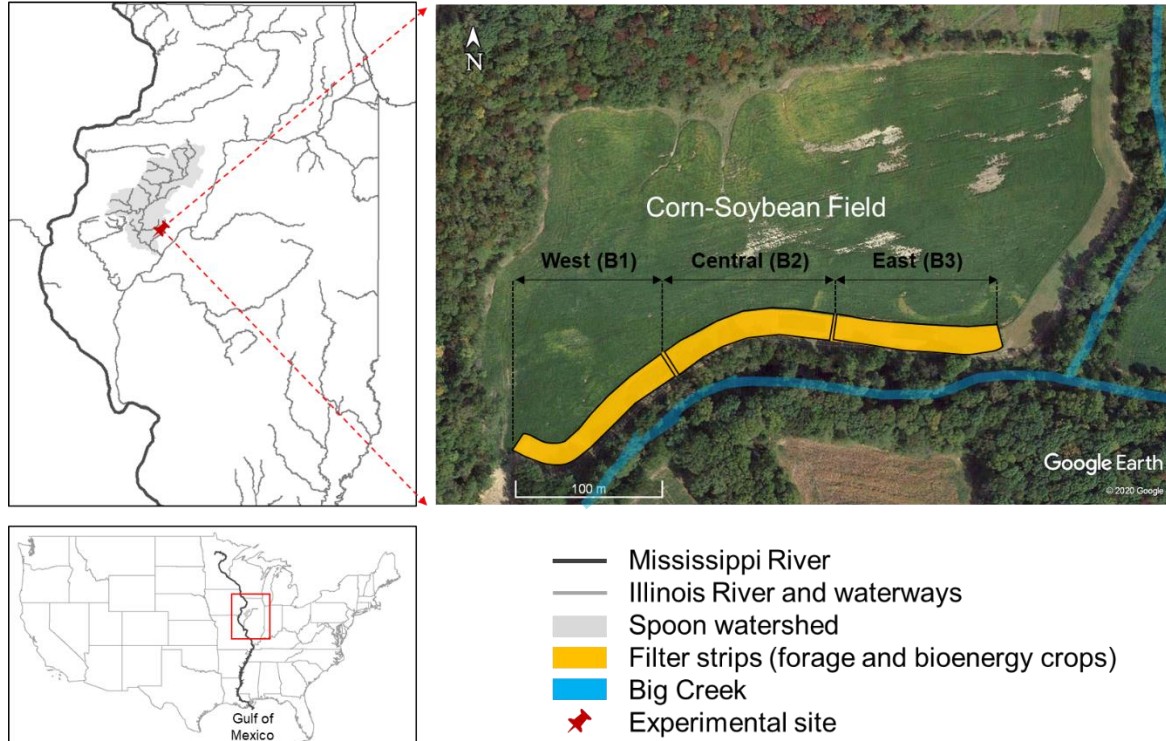

**Figure 1. The research site was in the Spoon watershed in Fulton County, Illinois (IL, 40°28'23" N 90°6'44" W). The perennial-based buffer strips were established at the edge of the farm field with continuous corn-soybean rotation from 2016 to 2019 (footnote: the letter B means blocks for the experimental design).**





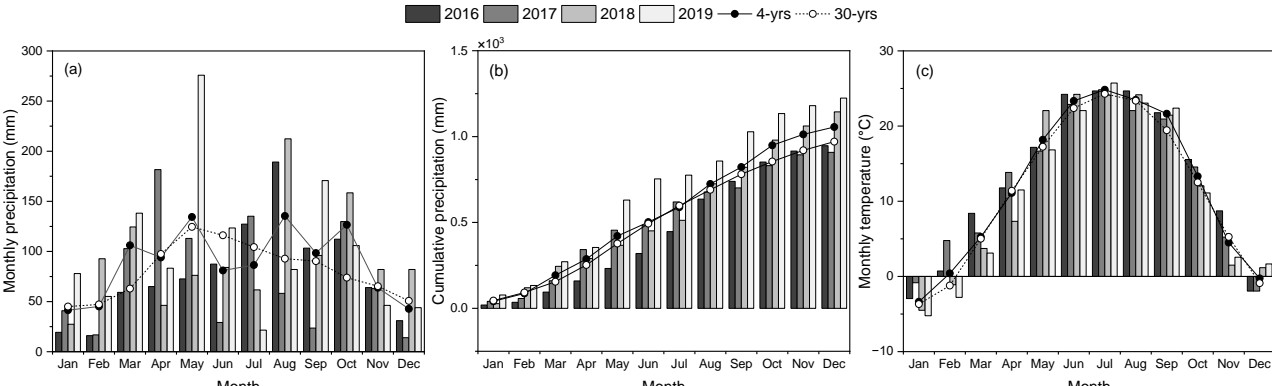


**Figure 2. Local weather conditions at the experimental site located in Fulton County, IL, across the four years of study (2016-2019), including (a) monthly and (b) cumulative precipitation and (c) monthly average temperature and the 30-year monthly average (1990-2019) (data: NOAA).**



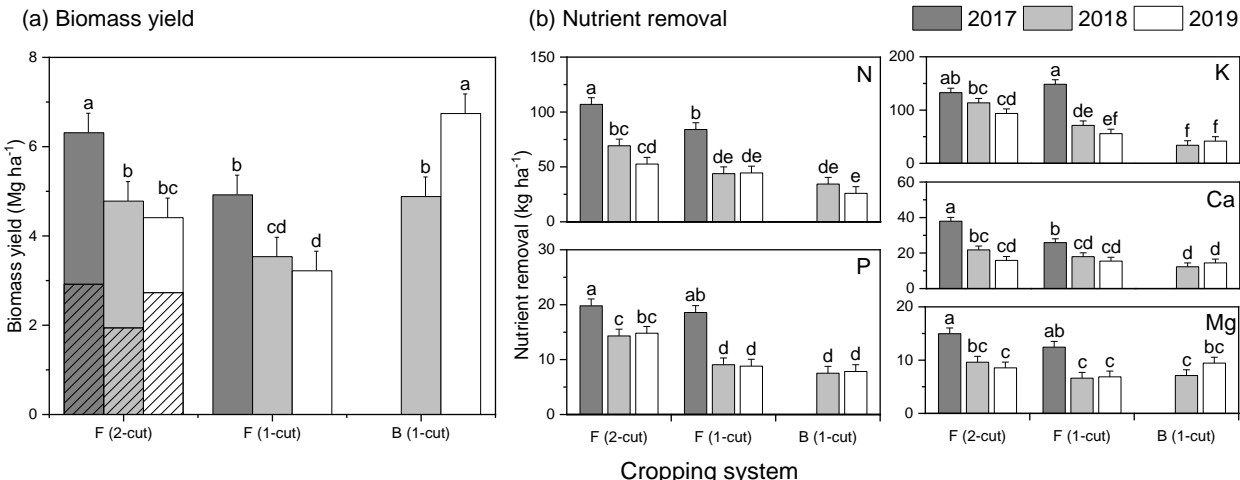

**Figure 3. Annual (a) biomass yields and (b) nutrient removal of forage and bioenergy feedstocks of the buffer strips, influenced by the harvest year and cropping system interaction (Y x CS) from 2017 to 2019. In the 2-cut forage system, the Summer- and Fall-harvested biomass were combined to evaluate the annual biomass yield (the summer-harvested yield was indicated by the spare pattern).**



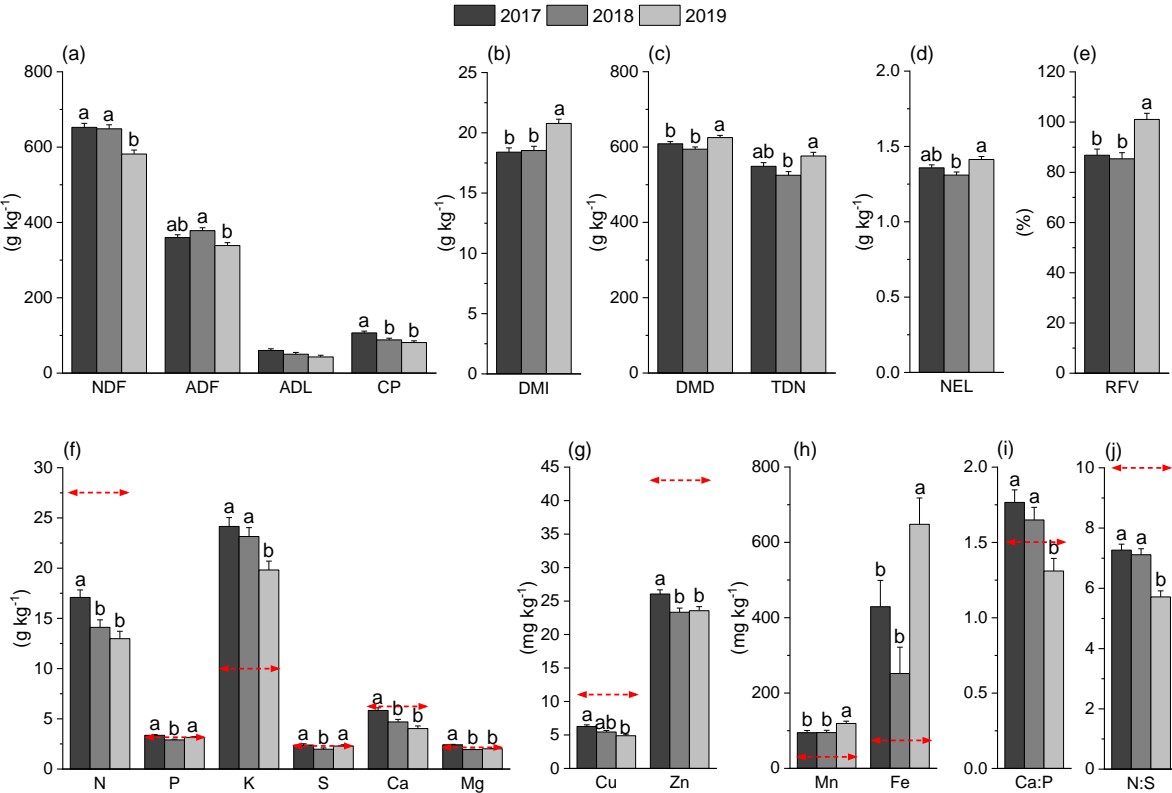

**Figure 4. The effect of cultivation year on feedstock quality of the harvested forage crops (averages of the 1- and 2-cut systems) from 2017 to 2019. The lowercase letters indicate mean separation (α=0.05), and no mean separations were applied if the variable effect was not significant. The red dashed line indicated the critical level of minerals required by dairy cattle (NRC, 2001).**





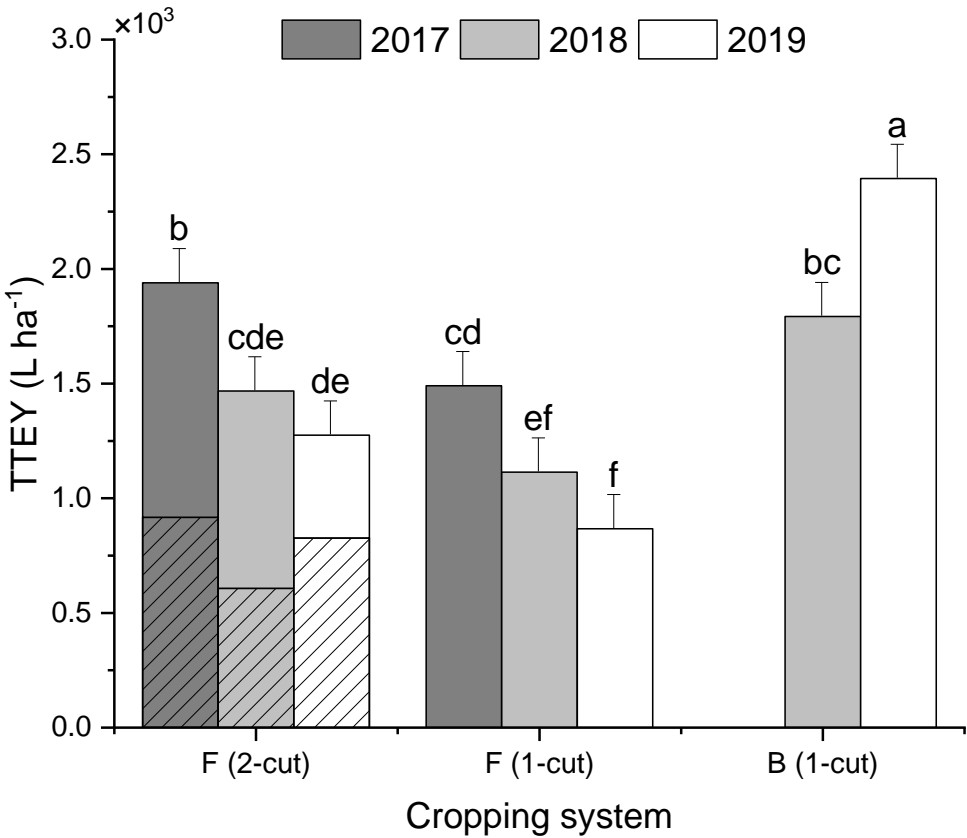

**Figure 5. The total theoretical ethanol yield (TTEY) of forage and bioenergy feedstocks, influenced by the harvest year and cropping**
**system interaction (Y x CS) from 2017 to 2019. In the 2-cut forage system, the Summer- and Fall-harvested biomass were combined**
**to evaluate the annual biomass yield (the summer-harvested yield was indicated by the spare pattern).**



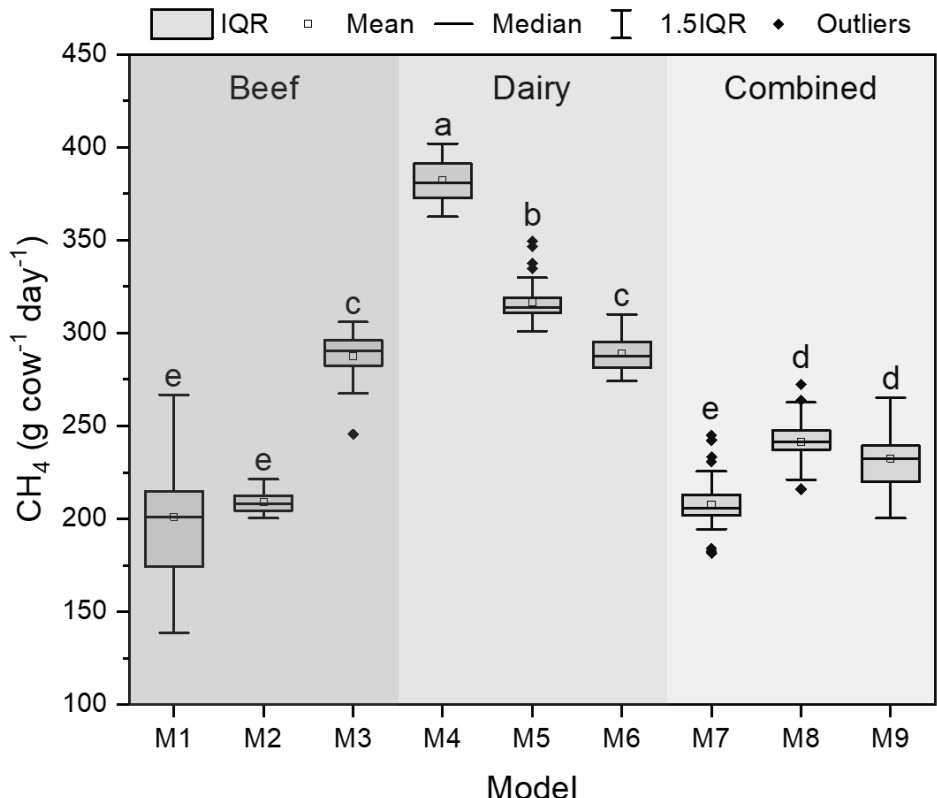

**Figure 6. Box plots of the predicted methane (CH₄) production based on forage nutritive quality using different prediction models (M1-M9). The predicted-CH₄ of forage feedstock was compared using the one-way ANOVA with a significance level of 0.05. The lowercase letters indicate mean separation organized from the highest to the lowest value for each column using the Tukey test.**



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
