# Peer review of "Biomass Yield Potential, Feedstock Quality, and Nutrient Removal of Perennial Buffer Strips under Continuous Zero Fertilizer Application"

_EGUsphere, 2024_

## Author Response (AR1)

**RC1 comments and response**

**General comments**

This experimental paper evaluates forage and bioenergy buffer strips in intensive cropping systems in relation to biomass yield, nutrient removal/uptake, and quality parameters related to forage and bioenergy feedstock. Overall, the paper is well-written well, well-motivated, and the results are thoroughly interpreted. The results and discussion lead to clear and useful information for optimal management of buffer strips considering competing objectives (e.g. optimal yield, nutrient removal, mitigation of GHGs and as substrate for livestock feed or bioenergy). The suggestions below include a few points needing addressed in the text and tables/figures as well as (minor) suggestions on writing style, discussion content, and typing errors.

Reply: The authors greatly appreciate the insightful suggestions and comments. The responses to each specific point raised are outlined below.

**Specific comments**

L95: Since you consider enteric CH4 but not other livestock GHGs, insert here a statement justifying why this emission was considered. E.g. you could provide data/references demonstrating it is the largest contributor to livestock GHG emissions.

Reply: Thanks for the suggestions. The following paragraph were added in the L95 of the introduction section to emphasize the contribution of enteric CH4 emissions. Furthermore, enteric methane (CH4) from ruminants constitutes a significant portion (approximately 32%) of global anthropogenic methane emissions, with cattle alone accounting for about 75% of these emissions from livestock. Since the production of methane is notably influenced by the intake and quality of feed, the last objective of this study was to employ empirical models to predict enteric CH4 production based on forage nutritive qualities.

L331: Under this section ('Feedstock quality') it would be good to indicate somewhere in the text what proportion forages make up in the livestock diets (i.e. as a proportion of total DMI). Since your considering only forage nutrient quality and not other (concentrate) feeds in the diet, any mineral deficiencies from forages may be offset by concentrate intake. As such it should be made clear that these results do not have conclusive impacts on overall diet quality of cattle. You could include somewhere in the discussion/conclusion that these results imply any mineral deficiencies/imbalances brought about by changes in forage nutrient quality should be considered in overall feed ration formulation.

Reply: Thank you for the suggestions and the following paragraph was added in the "feedstock quality" section to address this point. The dietary composition for cattle typically includes forage (e.g., hay, silage, and pasture) and concentrates (e.g., grains, protein sources, and other energy-dense feeds), supplemented with minerals and vitamins. The ratio of forage to concentrate ranged from 45:55 to 70:30 based on animal type, sex, target weight, and developmental stage (Aguerre et al., 2011; Briggs and Felix, 2021). For instance, forage constitutes approximately 50-70% of DMI to support proper rumen development during the growth phase, while in the finishing stage, forage usually reduced to about 30-40% to facilitate weight gain (Chen et al., 2015; Mialon et al., 2008). Although these results did not have conclusive impacts on overall diet quality of cattle, the gradual reductions in fiber (i.e., NDF) and nutrient (i.e., CP, Ca, Cu, Zn) concentrations, and nutrient ratio (i.e., Ca:P and N:S) provided valuable data for tailoring the forage to concentrate ratio to meet specific nutritional requirements and deficiencies in cattle. Additionally, an increasing number of cattle nutritionists and producers are adopting least-cost feed formulation strategies, which consider the total costs associated with diet mixing and daily feeding (Briggs and Felix, 2021). The minimal management effort applied in this study likely met with low-cost strategies for animal feed.

**Technical corrections (typing errors, etc.)**

*Introduction*

L40: Suggest take-up be replaced with 'uptake'.

Reply: Replaced with 'uptake'.

L59: Suggest reword 'maintain stand persistence' to for example 'sustain biomass yield' or something like this.

Reply: Rephrased this phrase with "sustain long-term biomass production".

L64: It may be helpful to clarify here whether you refer to the buffer strip or the main crop field.

Reply: Rephrased "polycultures" with "the perennial polyculture buffer strips".

L70: Replace 'not so much' with 'less so' (less colloquial/informal).

Reply: Fixed.

L82: 'Local economics'. This can be construed in many ways. Maybe replace with 'financial performance' if you are referring to for instance profit or income related indicators.

Reply: Rephrased "Local economics" with "financial performance".

L84: Didn't you calculate actual yield? This is different from yield potential. So maybe instead just write 'yield'.

Reply: Yes, we did measure the actual yield and changed the "yield potential" to "biomass yield".

*Methods*

L151: Should NDL be replaced with NDF. AS this is what is mentioned above (Neutral Detergent Fibre).

Reply: Thank you. It should be NDF and fixed.

L152: Dry matter intake cannot be included as an indicator of nutrient quality as it is instead an indicator of quantity consumed. Also, I suggest you state that this is predicted dry matter intake as your experiments do not actually calculate it (as shown in Table 1).

Reply: Changed the DMI from the quality index to the quantity index and emphasized that DMI was estimated using the biomass NDF concentration by adding the following sentence of "Dry Matter Intake (DMI) served as a quantitative index of animal feed to estimate the daily feed consumption per animal on a dry basis using the biomass NDF concentration." In table 5 and 6, the "quantity index" was also added.

Table 1: Are all these equations derived from Ameen et al. (2019)? It would be helpful to add this citation to the table heading or notes in addition to the text.

Reply: In Table 3, the description and calculation of indices were from Ball et al., (2017) and Ameen et al., (2019), which have been noted in Table 3.

L151: No need to capitalize 'Indices' in 'Quality Indices'.

Reply: Fixed.

L156: It would help if you provided a reference justifying how this equation was used to calculate TTEY.

Reply: The TTEY calculation was also based on Ameen et al., 2019, which was added in the text and Table 3.

L164: Often DMI is itself reported on a per unit animal basis. As such I suggest you clarify that your 'DMI' is in fact DMI per unit body weight. This makes it clearer why you need to multiply by body weight.

Reply: Thanks for the suggestions. The sentence was rephrased to "Although the DMI was commonly reported on a per unit animal basis, in this study, the dDMI was calculated by multiplying the DMI (expressed in per unit of body weight for comparisons across different animals and regions, $g\ kg^{-1}$) with the averaged body weight of cows across North America, Europe, Australia and New Zealand, which is 606 kg as reported by Appuhamy (2016)." to emphasize the unit of DMI is based on the body weight.

L169: This is a limitation then. You could potentially also omit the equations that use these variables.

Reply: Thank you for the suggestion. Although the daily metabolizable energy intake (MEi) and dietary fatty acid (FA) were considered as constant variables (MEi ~ 162 MJ d-1 and FA ~ 28 g kg-1) used in the equations #3, #4, and #5, the authors decided to keep these equations. Here are the reasons why the authors would like to keep them.

1) **Equations were equally selected from three cattle categories (Beef, Dairy, Beef and Dairy combined)** – Nine equations/models were selected based on their RMSPE and $R^2$ values from the published papers (Ellis et al., 2007; Nielsen et al., 2013) and equally selected from these three categories (i.e., equations #1-3 were trained by the beef dataset; equations #4-6 were based on the dairy dataset; equations #7-9 were based on the dataset of beef and dairy combined)

2) **Equation selections were based on different explanatory variables** – The enteric CH4 emissions can be predicted using the variables of DMI, ADF, ADL, CP, MEi, and FA alone or their combination. The rationale for the model selections in this study were to use different independent variables and their combination as much as we could. For example, although MEi was used as a constant in this study, the equation #3 (trained by beef dataset) included the explanatory variables of ADF and ADL which are different from the equation #1 and #2, which were also trained based on the beef dataset. Likewise, the FA was also used as a constant (used in equations #4 and #5), but the equations #4, #5, and #6 (trained by dairy dataset) used different explanatory variables for the CH4 predictions.

3) **More objective prediction results** – The CH4 predictions would be more objective by using the models trained by different dataset and explanatory variables.

L173: For harvest year, it would be better to specify actual individual years considered as factors (e.g. 2017, 2018, 2019) rather than listing the range (2017-2019).

Reply: Fixed.

*Results*

L189: 'TP'. For first use of abbreviation, spell out in full and put abbreviation in brackets.

Reply: Fixed.

L190: 'EC'. Same comment as above.

Reply: Fixed.

L192: Should this read 'the average across depths' (not locations)?

Reply: Here should be "the averages across locations". The phrase of "Although the soil EC (averages across depth) was similar in three sampling locations" showed that no location (B1, B2, and B3) effects on the overall soil EC (0-100 averages) were observed (B1: 0.20 dm m$^{-1}$; B2: 0.23 dm m$^{-1}$; B3: 0.23 dm m$^{-1}$). The soil depths, however, significantly influenced soil EC, so the averages across three locations (B1, B2, B3) showed that the top soil (0-5 cm: 0.52 dm m$^{-1}$; 5-10 cm: 0.33 dm m$^{-1}$) resulted in higher soil EC than the soil deeper than 10 cm (e.g., 10-20 cm: 0.16 dm m$^{-1}$; 20-30 cm: 0.11 dm m$^{-1}$; 30-60 cm: 0.10 dm m$^{-1}$; 60-100 cm: 0.10 dm m$^{-1}$).

L196: I presume you mean '30-year average precipitation' and not actually '30-year precipitation' (which would be the sum over 30 years)

Reply: Yes, here should be the "30-year average precipitation", which was fixed.

L217: Replace declined with declining. This reads better.

Reply: Fixed.

L222: 'were shown in Table 5'. Here you can use present tense.

Reply: Fixed.

L241: 'was declined' is poor English. Replace with either 'declined' or 'was declining'.

Reply: Changed to "declined"

L259: 'enteric CH productions'. Productions should be singular not plural (no 's').

Reply: Fixed.

*Discussion*

L281: 'SB' and 'SL'. If these are their first instances, spell out in full and put abbreviation in brackets. This applies to all abbreviated forage/grass species names in this section.

Reply: the full name and abbreviation have been addressed in the M&M section (L109-L118).

L282: PR-colver à PR-clover (typo).

Reply: Fixed.

L291: Suggest replacing 'decent' with 'substantial' (less subjective).

Reply: Fixed.

L296: I don't think 'referred' is the proper word here. Would 'implied' be what you intended here?

Reply: Here means "implied", so it is fixed.

L301: Suggest replacing 'inferred' with 'implied'.

Reply: Fixed.

L305: As with comment above, suggest replacing 'stand persistence' with 'yield sustainment' as this appears to better capture the meaning here (i.e. that the biomass yield remains higher for longer).

Reply: Changed the 'stand persistence' to 'yield sustainment'.

L306: The sentence starting on this line is too long and should be broken up.

Reply: The sentence of "In this study, the delayed harvest to the end of the growing season was implemented in both forage systems or until after a killing frost for the bioenergy system to improve feedstock stand longevity by providing extended time for vegetative development and reproductive tiller growth and to translocate nutrients to underground crop tissues that can be recycled for use in the following year" has been broken into two sentences.

L323: Suggest replacing 'nutrients moved down' with 'nutrient leaching/runoff'.

Reply: Fixed.

L341: It is unclear from this sentence what is meant by 'environmental/climatic' impact. I suggest you expand on this a bit to make it more clear what in particular you mean, e.g. that the climatic variables (temperature, rainfall) are outweighing the management factors in the experiments.

Reply: Two sentences were added for this clarification. In polyculture systems, the response of species compositions to management practices, along with biomass yield and chemical compositions, was significantly modulated by environmental variations such as weather and soil conditions (Cooney et al., 2023; Hong et al., 2014; Lin et al., 2023). In this study, the interaction between highly erodible soils and fluctuating precipitation levels likely intensified the complexities of the growth environment (e.g., inconsistent nutrient input via leaching and runoff), thereby diminishing the discernibility of experimental treatment effects.

L357: Suggest you add a reference for this P requirement for livestock, e.g. from NRC (2001).

Reply: Yes, the critical value of P in biomass is approximately 3.2 g kg-1 shown in NRC (2001). This reference was shown in Figure 4 and also added in the text.

L366: 'and is often required for supplementation'. I think this would be better re-worded as: 'and often necessitates supplementation' (clearer meaning).

Reply: Fixed.

L376: If you are reporting the range of N:S ratios (5.7~7.3) the dash (-) should be used in place of the tilde (~) (this is the correct way of reporting a range of values).

Reply: Fixed.

L376: Suggest replacing 'way' with 'substantially' as the former is colloquial.

Reply: Fixed.

L404: 'Oceanian' à 'Oceania' (typo)

Reply: Fixed.

**RC2 comments and response**

**General comments**

The evaluation of buffer strips as areas in intensive agriculture for forage and bioenergy production is an interesting and well-designed study. The analysis includes all relevant variables connected to yields, nutrient cycles and feed quality with thorough statistical assessments. The paper is well-written and deserves publication after minor revisions (see specific comments below).

Reply: Thank you all very much for valuable comments and suggestions. The author's comments and responses to RC2 are as follows:

**Specific comments**

L40: Please replace 'take-up' with 'uptake'.

Reply: Replaced with "uptake".

L59: Could you explain what is meant by 'maintain stand persistence' and find a better phrase for it?

Reply: Rephrased the phrase with "sustain long-term biomass production".

L82: What is meant by 'improving local economics'? Please clarify.

Reply: Rephrased "local economics" with "financial performance".

L85: 'Low-fertile marginal croplands' seems a double description. Please skip one adjective when it doesn't add information.

Reply: Removed "marginal" from this phrase.

Table 1: EC is not explained in caption

Reply: The explanation of soil EC was added in L195 while this term first occurred. In table 1, the full name of EC was added in footnotes.

L95: Please explain why livestock-related emissions are only represented by methane emissions.

Reply: The following paragraph was added at L95 in the introduction section to emphasize the contribution of enteric $CH_4$ emissions: "Furthermore, enteric methane (CH4) from ruminants constitutes a significant portion (approximately 32%) of global anthropogenic methane emissions, with cattle alone accounting for about 75% of these emissions from livestock (UNEP and CCAC report, 2021). Since the production of methane is notably influenced by the intake and quality of feed, the last objective of this study was to employ empirical models to predict enteric CH4 production based on forage nutritive qualities."

L190: Explain 'EC' at first occurrence.

Reply: The explanation of soil EC was added in L195. The rephrase sentence is "Soil electrical conductivity (EC) is used to measure the ability of soil water to conduct electricity, which is highly correlated with the soil nutrient concentrations (e.g., $NH_4^+$, $K^+$, $Na^+$, $NO_3^-$, $SO_4^{2-}$, and $Cl^-$, etc.), and often used for an alternative indicator for fertility."

L192: Please clarify the averaging levels, across locations or soil depths.

Reply: The phrase "Although the soil EC (averaged across depth) was similar in three sampling locations" indicated that no location effects (B1, B2, and B3) were observed on overall soil EC (0-100 cm averages) (B1: 0.20 dS m$^{-1}$; B2: 0.23 dS m$^{-1}$; B3: 0.23 dS m$^{-1}$). However, soil depth significantly influenced soil EC. The averages across the three locations (B1, B2, B3) showed that the topsoil (0-5 cm: 0.52 dS m$^{-1}$; 5-10 cm: 0.33 dS m$^{-1}$) had higher soil EC than soil deeper than 10 cm (e.g., 10-20 cm: 0.16 dS m$^{-1}$; 20-30 cm: 0.11 dS m$^{-1}$; 30-60 cm: 0.10 dS m$^{-1}$; 60-100 cm: 0.10 dS m$^{-1}$). The original sentence of "Although the soil EC (averages across depth) was similar in three sampling locations, the averages across locations showed that the EC in topsoil (0-10 cm) was around 3-4 times higher than deeper soil (<10 cm)." was rephrased to "Although the average soil EC across the 0-100 cm profile was similar at three sampling locations, the averages across locations showed that the EC in topsoil (0-10 cm) was around 3-4 times higher than in deeper soil (<10 cm)" for clarification.

L194: Could you add a short information about the seasonal precipitation cycle.

Reply: Yes, the following sentence was added in this section: The 30-year average indicated that the monthly precipitation in Illinois is well-distributed throughout the year, with substantial amounts occurring in mid-spring (April) and early fall (September). Compared to the 30-year average, the 4-year average precipitation displayed a more variable pattern, except for the generally low precipitation in winter (December to February). The cumulative data indicated that the precipitation in 2019 was substantially higher than in other years and the 30-year average precipitation.

L217: Please replace 'yearly declined trend' with 'declining annual trend'.

Reply: Fixed.

L222: Could you avoid sentences like this and better include an information from the table.

Reply: Rephrased the sentence to "The ANOVA analysis showed that the effects of harvest year, cropping system, and their interaction on feedstock chemical compositions, nutrient concentrations, and quality indices (Table 5)."

L249: Maybe better 'a similarly declining trend of the concentrations'.

Reply: Fixed. The new line# is L255.

L253: Could you describe the changes of the ratios without using percentages? I do have difficulties to compare ratios in this way.

Reply: The reductions of Ca:P and N:S rations were expressed as true values and the sentence was rephrased to "For nutrient ratios, Ca:P and N:S decreased substantially from 1.76 to 1.31 and 7.27 to 5.72, respectively, in the third harvest year, and both ratios were lower than the recommendation levels (the red dash lines shown in Fig. 4i and 4j)."

L295: I don't understand the word 'referred' here.

Reply: Changed to "implied".

L288: Here, you assume from the yield development about the decline of nutrients in the soil. Please also formulate it like this.

For L298, the sentence was rephrased to "In this study, the higher nutrient removal of the cool-season grasses resulted in possible depletion of soil nutrients over time, and the declined soil nutrients presumably accounted for a gradual decline in biomass yield in the forage system (approximately 0.9 Mg ha$^{-1}$ yr$^{-1}$ reduction) even though the yearly changes in soil fertility were not assessed."

L323: Please use a different formulation for 'moved down' such as e.g. 'taken off' or 'transferred from'.

Reply: Changed "moved down" to "For scavenging nutrients leaching/runoff from the C-S field,".

L331: In this section, methane emission equations are discussed depending on DMI. It would be beneficial to relate DMI and its quality to livestock production systems e.g. via a description of the feed baskets.

Reply: Thank you for the suggestions. I used the overall feed ration formulation to describe the "feed baskets" and the following paragraph was added in this section: "The dietary composition for cattle typically includes forage (e.g., hay, silage, and pasture) and concentrates (e.g., grains, protein sources, and other energy-dense feeds), supplemented with minerals and vitamins. The ratio of forage to concentrate ranged from 45:55 to 70:30 based on animal type, sex, target weight, and developmental stage (Aguerre et al., 2011; Briggs and Felix, 2021). For instance, forage constitutes approximately 50-70% of DMI to support proper rumen development during the growth phase, while in the finishing stage, forage usually reduced to about 30-40% to facilitate weight gain (Chen et al., 2015; Mialon et al., 2008). Although these results did not have conclusive impacts on overall diet quality of cattle, the gradual reductions in fiber (i.e., NDF) and nutrient (i.e., CP, Ca, Cu, Zn) concentrations, and nutrient ratio (i.e., Ca:P and N:S) provided valuable data for tailoring the forage to concentrate ratio to meet specific nutritional requirements and deficiencies in cattle. Additionally, an increasing number of cattle nutritionists and producers are adopting least-cost feed formulation strategies, which consider the total costs associated with diet mixing and daily feeding (Briggs and Felix, 2021). The minimal management effort applied in this study likely met with low-cost strategies for animal feed."

L340: Could you explain how the assumption 'presumably due to a substantial environmental/climatic impact' was derived?

Reply: This means that the complexities of the cultivation environment can be amplified by the interaction among different conditions, such as soil erosion, nutrient leaching, precipitation. The following sentences were added in this section: "In polyculture systems, the response of species compositions to management practices, along with biomass yield and chemical compositions, was significantly modulated by environmental variations such as weather and soil conditions (Cooney et al., 2023; Hong et al., 2014; Lin et al., 2023). In this study, the interaction between highly erodible zones and fluctuating precipitation levels likely intensified the complexities of the growth environment (e.g., inconsistent nutrient input via leaching and runoff), thereby diminishing the discernibility of experimental treatment effects."